# Investigating the Path Tracking Algorithm Based on BP Neural Network

**DOI:** 10.3390/s23094533

**Published:** 2023-05-06

**Authors:** Lu Liu, Mengyuan Xue, Nan Guo, Zilong Wang, Yuwei Wang, Qixing Tang

**Affiliations:** 1School of Engineering, Anhui Agricultural University, Hefei 230036, China; vliulu@ahau.edu.cn (L.L.);; 2Institute of Artificial Intelligence, Hefei Comprehensive National Science Centre, Hefei 230088, China; 3Hefei Institute of Technology Innovation Engineering, Chinese Academy of Sciences, Hefei 230094, China; 4Key Laboratory of Agricultural Sensors, Ministry of Agriculture and Rural Affairs, Hefei 230036, China

**Keywords:** automated vehicles, BP neural network, pure pursuit, look-ahead distance, path tracking

## Abstract

In this paper, we propose an adaptive path tracking algorithm based on the BP (back propagation) neural network to increase the performance of vehicle path tracking in different paths. Specifically, based on the kinematic model of the vehicle, the front wheel steering angle of the vehicle was derived with the PP (Pure Pursuit) algorithm, and related parameters affecting path tracking accuracy were analyzed. In the next step, BP neural networks were introduced and vehicle speed, radius of path curvature, and lateral error were used as inputs to train models. The output of the model was used as the control coefficient of the PP algorithm to improve the accuracy of the calculation of the front wheel steering angle, which is referred to as the BP–PP algorithm in this paper. As a final step, simulation experiments and real vehicle experiments are performed to verify the algorithm’s performance. Simulation experiments show that compared with the traditional path tracking algorithm, the average tracking error of BP–PP algorithm is reduced by 0.025 m when traveling at a speed of 3 m/s on a straight path, and the average tracking error is reduced by 0.27 m, 0.42 m, and 0.67 m, respectively, at a speed of 1.5 m/s with a curvature radius of 6.8 m, 5.5 m, and 4.5 m, respectively. In the real vehicle experiment, an electric patrol vehicle with an autonomous tracking function was used as the experimental platform. The average tracking error was reduced by 0.1 m and 0.086 m on a rectangular road and a large curvature road, respectively. Experimental results show that the proposed algorithm performs well in both simulation and actual scenarios, improves the accuracy of path tracking, and enhances the robustness of the system. Moreover, facing paths with changes in road curvature, the BP–PP algorithm achieved significant improvement and demonstrated great robustness. In conclusion, the proposed BP–PP algorithm reduced the interference of nonlinear factors on the system and did not require complex calculations. Furthermore, the proposed algorithm has been applied to the autonomous driving patrol vehicle in the park and achieved good results.

## 1. Introduction

In recent years, substantial progress has been made in the development of autonomous vehicles [1,2,3]. Among them, high-precision path tracking algorithms are an important component in achieving autonomous driving [4]. Path tracking algorithms enable vehicles to reach specified locations along a planned path. Generally speaking, path tracking algorithms can be divided into model-based methods and geometric methods, both of which have been widely applied in vehicle path tracking.

Model-based path tracking methods consider the kinematic or dynamic characteristics of the vehicle to predict and compensate for potential errors between the vehicle’s motion and the intended path. The methods consist of Linear Quadratic Regulator (LQR) [5,6,7,8], fuzzy control [9], and Model Predictive Control (MPC) [10,11]. Although model-based methods have good accuracy, they are sensitive to the accuracy of localization and path curvature, as well as model accuracy and parameters. In addition, they require a large amount of computational resources and a longer time to calculate feasible solutions [12]. Geometric-based path tracking methods use the geometric relationship between the vehicle and the path, making them less affected by path smoothness, missing waypoints, and localization errors. The two main geometric-based algorithms are the pure pursuit (PP) algorithm [13] and the Stanley algorithm [14,15]. Among them, the PP algorithm is a classic path tracking algorithm. This method selects lookahead points on a reference trajectory and constructs geometric relationships between the vehicle’s rear axle center and the lookahead point, which solves the front wheel steering angle that causes the rear wheel center to pass through the reference point [16]. Compared to the Stanley method, the PP algorithm has better robustness for non-smooth paths and large heading errors. Additionally, this method has a simple implementation and is the most widely used and popular algorithm for autonomous vehicles [17,18].

The objective of this study is to investigate the path tracking problem in park scenarios, which involves scenarios with obstacle avoidance, lane changing, and continuous changes in path curvature. In these environments, the vehicle’s driving speed is low, the vehicle’s positioning information may have significant errors, and the path to be tracked exists in the form of non-smooth waypoints. In these situations, geometric path tracking methods are usually superior to model-based path tracking methods [19]. Therefore, the work presented in this paper is actually an improvement on the PP algorithm to achieve more accurate path tracking in park scenarios.

The traditional PP algorithm is limited by the lookahead distance. Firstly, the establishment of geometric relationships involves the selection of lookahead points. If the lookahead distance is increased, the stability of the vehicle increases, but the tracking accuracy decreases. On the other hand, if the lookahead distance is decreased, the tracking accuracy improves but the vehicle may experience oscillations. To address this issue, Qin Peng et al. used an adaptive strategy to adjust the lookahead distance, which allowed the vehicle to converge and track the trajectory without heading oscillations, but the convergence speed was not maximized [20]. Zhang et al. used the IATE optimization criterion to simulate the optimal lookahead distance based on a Kalman filter and a PP model. Although this study improved path tracking capability, it introduced various complex functions and reduced system robustness [21]. Considering path curvature, heading error, vehicle speed, and lateral error, dynamically adjusting the lookahead distance can also effectively improve path tracking accuracy [22,23]. On the other hand, at locations where the path curvature changes, the curvature at the lookahead point may differ from the curvature at the vehicle’s rear axle center, causing corner-cutting problems and resulting in cross-track errors. Andersen et al. proposed selecting lookahead points with a constant offset distance around the path to address the corner-cutting issue. By setting the offset distance appropriately, the corner-cutting issue can be avoided at certain curvatures, but there may be offset errors at other curvatures [24]. Ahn et al. proposed a heuristic method to select lookahead points based on the relationship between the vehicle and the path, and this method ensures tracking accuracy while avoiding the problem of corner cutting, but when there are significant changes in path curvature, the sudden change in lookahead distance can cause oscillations and reduce smoothness [25]. In summary, researchers have conducted extensive studies on the lookahead distance of PP models. However, the above methods require a large number of experiments to determine their key parameters, and the real-time performance of the control algorithm needs to be improved. Moreover, the path tracking of autonomous vehicles in industrial parks needs to consider the disturbance caused by various factors, which cannot be described by a simple linear model [26].

Therefore, in this paper, we introduce the use of BP neural network to optimize the PP algorithm. Firstly, the calculation process of the vehicle’s steering angle based on the pure pursuit algorithm is derived. By recording the vehicle speed, road curvature radius, and lateral error under multiple path scenarios, a dataset is then obtained as input to train a BP neural network model. The obtained correction coefficients through the model are used to optimize the front wheel steering angle values calculated by the pure pursuit algorithm in order to achieve better path tracking accuracy in the park. Finally, the effectiveness of the algorithm is verified through simulations and real vehicle experiments. The experimental results show that compared with traditional geometry-based tracking methods, this method can effectively adjust the position and posture of the vehicle. Moreover, when there are significant changes in path curvature, the path tracking accuracy is greatly improved. This method provides a new idea for the study of path tracking in industrial parks and has been applied in practice on autonomous vehicles in industrial parks.

## 2. Materials and Methods

The control object in this paper is an electric patrol vehicle (AW6042K, ALWAYZ, Chuzhou, China) equipped with 16-line lidar (RS-16, RoboSense, Shenzhen, China), a Bynav A1-series high-precision navigation board (A1-3, Bynav, Changsha, China), and a bottom controller for the vehicle. The overview of the autonomous vehicle and its sensors is shown in Figure 1 and the parameters are provided in Table 1.

To obtain the vehicle position, Bynav A1 GNSS/INS integrated navigation board is usually used in the automatic navigation control system. It adopts deep coupling technology combining Global Navigation Satellite System (GNSS) positioning and Inertial Measurement Unit (IMU) [27], where the IMU can compensate for the shortcomings of the Global Positioning System (GPS) in the area where the satellite signal is obscured, and the positioning accuracy is low. Meanwhile, the GPS signal can correct the IMU’s accumulated error [28,29].

An overview of the entire system can be seen in Figure 2. First, the GPS and IMU modules provide the vehicle’s position and speed information. After that, calculate the lookahead distance (LD) based on the vehicle’s speed. In the meantime, the trained neural network model solves the optimal control coefficient K according to the current vehicle speed, path curvature, and lateral error, which is used to correct the front wheel steering angle of the vehicle determined by the PP algorithm. Finally, the corrected steering angle of the front wheel is sent to the actuator to realize the automatic steering of the vehicle.

### 2.1. Vehicle Kinematic Models

The main hypothesis in the kinematic analysis with a two-wheel model is that the vehicle is running on a smooth road, where the tires and the ground only generate longitudinal pressure. To simplify the calculations, the vehicle model can be simplified to a two-wheel model, called the Ackerman model [26]. Figure 3 depicts the kinematic analysis of the simplified vehicle mode, where $\left(x_{f},y_{f}\right)$ and $\left(x_{r},y_{r}\right)$ are the axes of the front and rear wheels, respectively, and L is the vehicle’s wheelbase. In all calculations, the longitudinal direction of the vehicle is considered as the reference and deviations to the right and left are assumed positive and negative, respectively. δ is the front wheel steering angle and θ is the heading angle of the vehicle, which is the angle from the longitudinal counterclockwise rotation of the vehicle about the *y*-axis, and the value range is 0–360°.

According to the established two-wheel model, the front wheel steering angle is defined as follows:(1)$$\delta =\arctan\frac{L}{R}$$

The kinematics model can be obtained using the geometric constraints of the front and rear wheels:(2)$$\left\{\begin{array}{l} x^{\prime }\left(t\right)=v\cdot \sin\theta \\ y^{\prime }\left(t\right)=v\cdot \cos\theta \\ \theta^{\prime }\left(t\right)=v\cdot \tan\theta /L \end{array}\right.$$
where v and $\theta^{\prime }(t)$ are the longitudinal and angular velocity of the vehicle, respectively. Moreover, $x^{\prime }(t)$ and $y^{\prime }(t)$ denote the vehicle partial velocity along the *x*-axis and *y*-axis, respectively.

### 2.2. Analysis of Pure Pursuit Algorithm of The Vehicle

The PP algorithm is a geometric method to find the optimum arc passing through the current position and the preview point [30]. Figure 4 shows the schematic diagram of this algorithm. The look-ahead point of the vehicle body on the reference path is $\left(x_{t},y_{t}\right)$. The turning radius, and the distance from the center of the vehicle body to the preview point are R and $L_{d}$, respectively. The angle between the vehicle body and the connection between the preview point and the center point of the rear axle is α.

According to the vehicle kinematics model, the steering angle of the front wheel can be expressed as:(3)$$\frac{L_{d}}{\sin\left(2\alpha \right)}=\frac{R}{\sin\left(\frac{\pi }{2}-\alpha \right)}$$
(4)$$\delta =\arctan\frac{2L\sin\alpha }{L_{d}}$$
where δ is the front wheel steering angle of the vehicle along the curve to the preview point.

Look-ahead distance is a key parameter affecting vehicle tracking performance in PP model, which is generally described as a first-order function related to vehicle speed, and can be calculated by using the following expression:(5)$$L_{d}=kv+L_{\min}$$
where $L_{\min}$ is the minimum of the look-ahead distance and k is the control coefficient for the vehicle speed.

The work presented in this paper takes into account the non-linear disturbances caused by vehicle speed, path curvature radius, and lateral error on path tracking based on PP algorithm. The BP neural network model is used to obtain real-time control coefficients to correct the steering angle of the front wheels. The aim is to improve the accuracy of path tracking. The formula is expressed as follows:(6)$$\delta_{c}=K_{c}\arctan\frac{2L\sin\alpha }{L_{d}}$$
where $K_{c}$ represents the control coefficient output by the BP neural network model in the subsequent work. $\delta_{c}$ represents the steering angle of the front wheels after being corrected by the control coefficients.

## 3. Steering Angle Correction Method

### 3.1. Data Set

The dataset in this paper contains 140×3 samples, including data from three categories: vehicle speed, lateral error, and path curvature radius. Vehicle speed is obtained from the GNSS/IMU module, and as shown in Equation (5), there is a linear relationship between vehicle speed and forward distance, which is an important factor in adjusting vehicle steering angle. Similarly, the lateral error and path curvature radius have a non-linear relationship with the vehicle steering angle, which has a certain impact on the final tracking accuracy. The calculation process for the lateral error and path curvature radius is given as follows.

For the lateral error, first, the matching point $\left(x_{m},y_{m}\right)$ is determined on the reference path, and then two points, $\left(x_{m-1},y_{m-1}\right)$ and $\left(x_{m+1},y_{m+1}\right)$, are determined before and after the matching point, respectively. A straight line is formed by connecting these two points, and its equation is expressed as follows:(7)Ax+By+C=0
where *A*, *B*, and *C* are the coefficients in the straight line.

Using point $\left(x_{m-1},y_{m-1}\right)$ as the origin, $\left(x_{m+1},y_{m+1}\right)$ and the rear axle center coordinate $\left(x_{r},y_{r}\right)$ of the vehicle (as shown in Figure 2) as the endpoints, two vectors ${\vec{n}}_{1},{\vec{n}}_{2}$ are constructed, respectively. The lateral error e is then represented as:(8)$$e=-sign({\vec{n}}_{1}\times {\vec{n}}_{2})\cdot \frac{Ax_{r}+By_{r}+C}{\sqrt{A^{2}+B^{2}}}$$

The direction of the lateral error is determined by the cross-product of the vectors ${\vec{n}}_{1},{\vec{n}}_{2}$.

The calculation of the path curvature radius is actually the result of traversing all the reference waypoints within a 15-m range in front of the vehicle. Assuming the number of reference path points is N, for the i-th point $\left(x_{i},y_{i}\right)$, an external circle is constructed by combining the previous and subsequent points $\left(x_{i-1},y_{i-1}\right)$ and $\left(x_{i+1},y_{i+1}\right)$, and the radius of the external circle $r_{i}$ is considered as the curvature radius of the path at this point. The curvature radius of the entire path is represented as follows:(9)$$r=\min{\left.\left(r_{i}\right)\right|}_{i=1}^{N}$$

The collected data information is shown in Figure 5, with a speed range of 0.5 to 2.0 m/s, a path curvature radius range of 2.5 to 11.0 m, and a lateral error range of 0.2 to 0.6 m. To avoid the impact of differences in data scale on network learning, the input data is normalized to between $\left[-1,1\right]$. The normalization formula is as follows:(10)$$X^{\prime }=\left(1-\left(-1\right)\right)\cdot \frac{X-X_{\min}}{X_{\max}-X_{\min}}+1$$

### 3.2. Neural Network Model

BP neural network is a multilayer feedforward neural network trained with the error back propagation algorithm. In terms of structure, the BP neural network consists of an input layer, a hidden layer, and an output layer. Two processes are involved in the learning process of the neural network, the forward propagation of the signal FP (loss), and the back propagation of the error BP (error back propagation) [31,32]. The initial weight is set randomly and then modified in back propagation according to the difference between the training data (i.e., the true value during training) and the output data (i.e., the actual value that is planned to achieve during training). Repeat the forward and backward propagation processes until the difference between the training data and the output data meets the training accuracy (i.e., the maximum acceptable average error between the training data and the actual data). In the standard BP neural network, the gradient descent algorithm is used, and the network weight is adjusted backward along the gradient of the performance function. As a result, the predicted value approaches the actual value continuously. Consequently, it can directly improve vehicle tracking accuracy.

The topological structure of the BP neural network is shown in Figure 6. The forward propagation process is constructed from the input layer first. For the input to the l-th layer of a neural network, the following formula applies:(11)$$z^{l}=w^{l}\alpha^{l-1}+b^{l}$$
where $w^{l}$ is the weight matrix from the l−1-th layer to the l-th layer, $\alpha^{l-1}$ is the output of the neurons in the l−1-th layer, and $b^{l}$ is the bias vector from the l−1-th layer to the l-th layer.

The output of the l-th layer of the neural network is represented as:(12)$$\alpha^{l}=f_{l}\left(z^{l}\right)$$
where $f_{l}\left(z^{l}\right)$ is the activation function of the l-th layer.

The activation function of hidden layer neurons can be defined as follows:(13)$$f_{1}(x)=\frac{e^{x}-e^{-x}}{e^{x}+e^{-x}},f_{1}(x)\in (0,1)$$

On the other hand, a linear transfer function is used as the activation function of the output layer neuron. This function can be expressed as follows:(14)$$f_{2}(x)=x$$

The basic principle of the backpropagation algorithm is to calculate the error signal and propagate it layer-by-layer to calculate the partial derivative of each weight on the whole network error function. Eventually, we can use the gradient descent algorithm to minimize the loss function and optimize the weights and biases of the neural network. For linear regression problems, this article adopts MSE as the loss function, which is expressed as follows:(15)$$E\left(w,b\right)=\frac{1}{2}\sum_{i=1}^{N}{\left(y_{i}-{\hat{y}}_{i}\right)}^{2}$$
where $y_{i}$ represents the i-th predicted result of the model and ${\hat{y}}_{i}$ represents the true value of the i-th instance.

According to backpropagation and chain rule, the error signal $\delta^{l}$ of the l-th layer can be calculated from the error value of the l+1-th layer.
(16)$$\delta^{l}=\frac{\partial E}{\partial z^{l}}=\frac{\partial E}{\partial z^{l+1}}\frac{\partial z^{l+1}}{\partial \alpha^{l}}\frac{\partial \alpha^{l}}{\partial z^{l}}={\left(w^{l+1}\right)}^{T}\delta^{l+1}⊙f_{l}{}^{\prime }\left(z^{l}\right)$$
where $f_{l}{}^{\prime }\left(z^{l}\right)$ is the reciprocal of $f_{l}\left(z^{l}\right)$ and ⊙ is the Hadamard product, which represents the element-wise multiplication.

The gradient expressions and update values for $w^{l}$ and $b^{l}$ in the BP neural network can be represented as follows:(17)$$\left\{\begin{array}{l} \frac{\partial E}{\partial w^{l}}=\frac{\partial E}{\partial z^{l}}\frac{\partial z^{l}}{\partial w^{l}}=\delta^{l}\alpha^{l-1}\\ \frac{\partial E}{\partial b^{l}}=\frac{\partial E}{\partial z^{l}}\frac{\partial z^{l}}{\partial b^{l}}=\delta^{l} \end{array}\right.$$
(18)$$\left\{\begin{array}{l} \Delta w^{l}=-\eta \delta^{l}\alpha^{l-1}\\ \Delta b^{l}=-\eta \delta^{l} \end{array}\right.$$
where η represents the learning rate.

After determining the structure of the BP neural network, weights and biases are randomly generated, and the sample data is input to the model to obtain predicted values through nonlinear transformation by activation functions. The error backpropagation algorithm updates weights and biases based on the difference between predicted and true values, aiming to minimize the model error.

### 3.3. Model Parameters

In the BP neural network, some hyperparameters need to be manually set. The number of hidden layer nodes has a direct impact on the final training effect of the model. Too few nodes in the hidden layer can lead to increased prediction error in the final results, while too many nodes can lead to overfitting and the model being trapped in local optimization. In this paper, the range of hidden layer node numbers is determined based on empirical formulas [33,34]:(19)$$G<\sqrt{\left(m+n\right)}+a$$
where G, m, and n are the number of nodes in the hidden, output, and input layers, respectively.a is an integer that varies in the range $\left[0,10\right]$.

According to Formula (21), the range of the number of hidden layer nodes is determined to be $\left[2,12\right]$. Different numbers of hidden layer nodes are set within this range, and the dataset is used to train the model 10 times. The prediction accuracy on the test set is then calculated and compared for different numbers of hidden layer nodes [35]. As shown in Figure 7, the MSE gradually decreases as the number of hidden layer nodes increases in the range of $\left[0,14\right]$. When the number of nodes is 10, the MSE basically converges. Therefore, in this study, the number of hidden layer nodes is set to 10.

The sample data is randomly divided into a training set, a validation set, and a test set, with a ratio of 70%, 15%, and 15% respectively. In addition, the training iterations, target error, learning rate, and validation check value of the model are set to 1000, $1.0\times 10^{-6}$, and 0.05, respectively.

### 3.4. Evaluation of Neural Network Training

The training set of neural networks is obtained by setting up curves of different curvatures, using different speed and lateral error estimates, recording the control coefficient with minimum relative tracking errors, and obtaining 100 sets of training data. Among the input parameters of the training set are the vehicle speed, the radius of curvature, and the lateral error, and the output parameter is the control coefficient. The relationship between input and output is shown in Figure 5.

To evaluate the accuracy of the training results, the mean square error and the correlation coefficient are used as evaluation indicators. Figure 6 reveals that the smaller the MSE, the greater the prediction accuracy, and the closer the correlation coefficient R to 1 [36].

Figure 8 indicates that the minimum mean square error and the corresponding correlation coefficient R at the 8th iteration are $2.0511\times 10^{-3}$ and 0.99637, respectively. Figure 9 shows the distribution of the coefficient error obtained from the training data. The predicted value of the output coefficient is compared with the experiment. The error mainly varies within the range (−0.01, 0.01) and the average error is 0.0048. The results reveal that the error is relatively small, and the model meets the accuracy requirements.

## 4. Simulation Experiments

### 4.1. Straight Path Tracking

The initial position of the vehicle is set to (0, −2), which is 2 m away from the starting point of the preset path. The linear velocity, the speed coefficient, and the minimum look-ahead distance are set to 3 m/s, 0.1, and 1.75 m, respectively. Moreover, the simulation step is set to 0.1 s, and the tracked straight line is y=x. The performance of the BP–PP algorithm is then compared with that of the PP algorithm and the Stanley algorithm. The straight path tracking of the three methods and the lateral error of the three tracking methods are presented in Figure 10 and Figure 11, respectively.

The obtained results show that the BP–PP algorithm has the fastest response speed. When the initial error is 2 m, the vehicle can track the path stably for about 5 m. After stable tracking, the average tracking error of the PP algorithm, Stanley algorithm, and BP–PP algorithm on the straight path is 0.038 m, 0.049 m, and 0.013 m, respectively.

### 4.2. Curve Path Tracking

Furthermore, a complete path composed of three paths with different curvatures is set up to compare the robustness of the algorithm on the curved path. The average curvature radius of the three paths is 6.8 m, 5.5 m, and 4.5 m, respectively. Meanwhile, the vehicle speed is set to 1.5 m/s, and the initial position is set to (0, −2). The performance of the BP–PP algorithm is compared with that of the PP algorithm and Stanley algorithm. The tracking effect of the three methods is compared. The three methods’ tracking conditions and tracking errors on different corners are presented in Figure 12 and Figure 13, respectively.

The performed analyses show that for an initial error of 2 m, the BP–PP algorithm has the fastest response speed, and the vehicle can quickly track the path. The obtained results are presented in Table 2. It is observed that when three curves with different curvature radii are set, the BP–PP algorithm effectively reduces the tracking error of the vehicle compared with the PP algorithm and Stanley algorithm, indicating the superiority of the proposed control algorithm.

## 5. Experiments

### 5.1. Experimental Scene

The experiment was conducted on two two-way lanes with a width of 7 m. The area enclosed by the road was approximately rectangular, with a unilateral side length of about 100 m. The road surface is asphalt, moderately hard, and there is a speed bump at the bend. Figure 14a describes the tracking scene of the vehicle in a right-angle curve, Figure 14b describes the tracking scene of the vehicle in a straight line, and Figure 14c describes the tracking scene of the vehicle in an arched curve. It is worth noting that the GNSS module must be working in RTK mode before experimenting.

### 5.2. Rectangular Path Tracking Experiment

Simulation experiments demonstrate that the proposed path tracking algorithm is adaptable and robust. In order to evaluate the proposed method further, a road surface experiment is carried out. The rectangular path consists of a right-angle curve and a straight path. The initial tracking error is set to 0.25 m, and the deceleration belt is set in the middle as interference. Comparative experiments are conducted with PP and BP–PP algorithms. GPS data of the desired trajectory and the tracking trajectory are put in a coordinate system, and the obtained tracking results are presented in Figure 15. By comparing the automatic navigation route data to the predefined route data, we can estimate the lateral error of the path tracking.

The tracking error is calculated by comparing the recorded GPS data with the target path after the vehicle has achieved stable tracking. The obtained results in this regard are shown in Figure 16. The analysis demonstrates that the vehicle can quickly track the path for an initial error of −0.24 m. Moreover, Table 3 indicates that the deceleration bumps do not affect the tracking results for tracking on a rectangular curve. It is inferred that compared with the conventional PP algorithm, the BP–PP algorithm effectively reduces the vehicle’s tracking error on curved and straight paths, thereby reflecting the superiority of the proposed control method.

### 5.3. Large Curvature Path Tracking Experiment

Experiments on rectangular paths have demonstrated that the proposed BP–PP algorithm achieves significant results. To further verify that the proposed algorithm was adaptable to the environment, a large curvature path tracking experiment was designed. We set a target path with an average curvature radius of 6.5 m and the initial lateral error of the vehicle was 0.24 m. PP algorithm and BP–PP algorithm were used for path tracking respectively. Similarly, the GPS data of vehicle automatic tracking trajectory is recorded, and the recorded vehicle tracking trajectory is compared with the target path to obtain the tracking lateral error. The path data is shown in Figure 17.

Statistics are presented in Table 4 and the results of the lateral error calculation are shown in Figure 18. Results indicate that the proposed BP–PP path tracking algorithm is still capable of tracking paths with large curvatures. As a result of this algorithm, the target path can be tracked quickly and vehicle tracking errors can be effectively reduced.

The on-path testing results indicate that the tracking accuracy of the vehicle slightly decreased compared to the simulation experiments. However, compared to the traditional geometric-based path tracking methods, the proposed BP–PP algorithm exhibited higher accuracy and robustness in both simulation and on-road testing. The proposed method effectively reduces path tracking errors and exhibits fast response performance on paths with low curvature. When facing the potential corner-cutting problem caused by changes in path curvature, the proposed method can effectively reduce cross-track errors. In the simulation experiments, it was evident that the tracking results of the BP–PP algorithm were closer to the desired path, and the tracking performance of the algorithm improved more as the road curvature increased. This result was also demonstrated in on-road testing with high-curvature paths, where the BP–PP algorithm exhibited better tracking performance on curved paths. In summary, the proposed BP–PP algorithm achieved good tracking performance on paths with curvature changes in the park, demonstrating its innovation and practical application in autonomous vehicles in the park.

## 6. Conclusions

This paper presents a path tracking algorithm based on BP neural networks. To improve the accuracy of the front wheel steering angle, the BP neural network is used as the basis for the PP algorithm. Based on simulations and real vehicle experiments, it has been shown that the proposed algorithm can improve the tracking accuracy of the vehicle on straight lines and curves, and can adjust control coefficients automatically based on road conditions and vehicle status. As a result, the overall accuracy of the system is higher and the robustness is greater. This shows that it is effective and feasible to introduce machine learning into the traditional trajectory tracking algorithm to enhance the performance of the system. Despite the better stability, we also noticed that the vehicle kinematics methods had a limited increase in accuracy, meaning that we might need to use a theory-based path tracking algorithm to further optimize the system. Although this paper gives three main parameters that affect path tracking accuracy, in practice, more factors may alter path tracking accuracy.

Therefore, we will focus our future research on the following aspects to improve path tracking accuracy. As a first step, instead of algorithms based on kinematic models, we will use theory-based algorithms, such as MPC or LQR. After that, we will consider more complex working environments such as farmland, hills, and other environments that impact path tracking more. Lastly, we hope to introduce deep learning into the system in future research, and to deep-dive into the factors that affect the accuracy of path tracking, and this will also be the goal of our future work.

## Figures and Tables

**Figure 1 sensors-23-04533-f001:**
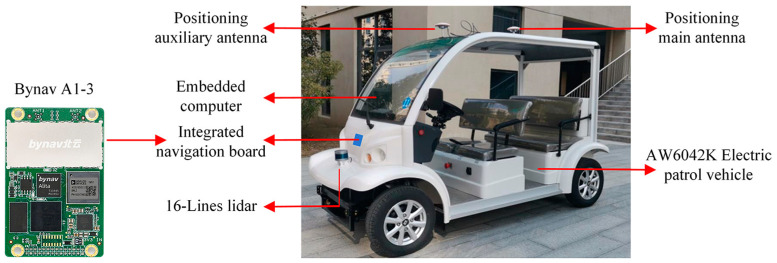
Automatic tracking electric patrol vehicle platform and related equipment mounted thereon.

**Figure 2 sensors-23-04533-f002:**
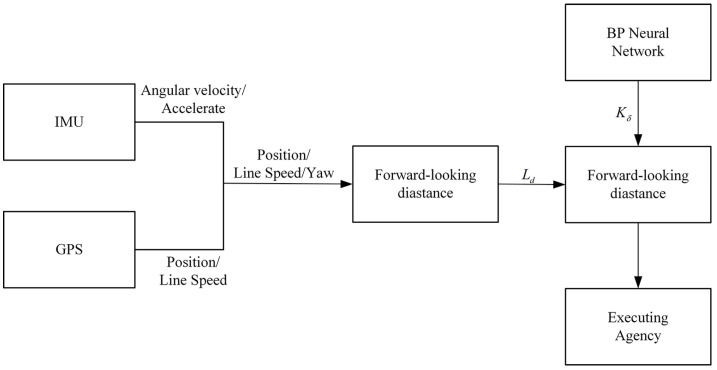
Schematic diagram of the path tracking system.

**Figure 3 sensors-23-04533-f003:**
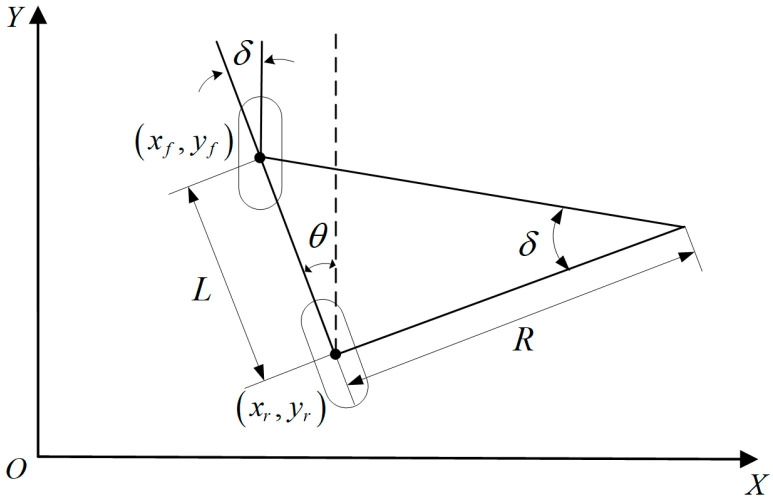
Schematic diagram of the two-wheel model.

**Figure 4 sensors-23-04533-f004:**
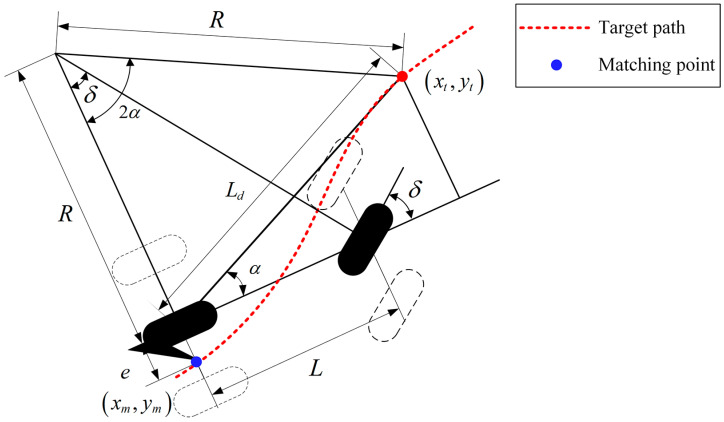
Schematic diagram of the PP algorithm geometric model.

**Figure 5 sensors-23-04533-f005:**
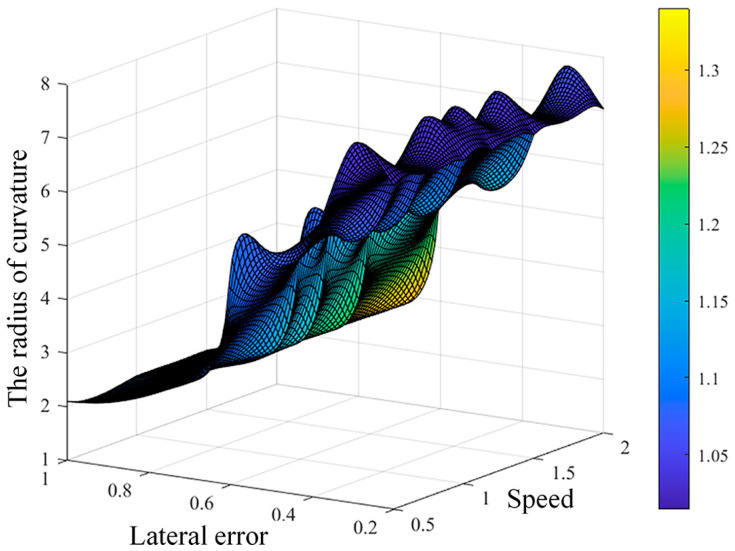
Data range schematic of dataset.

**Figure 6 sensors-23-04533-f006:**
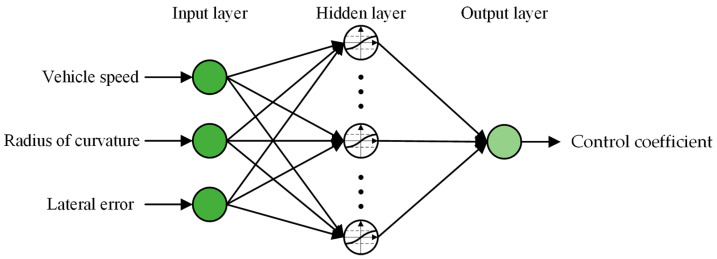
Topology structure of BP neural network.

**Figure 7 sensors-23-04533-f007:**
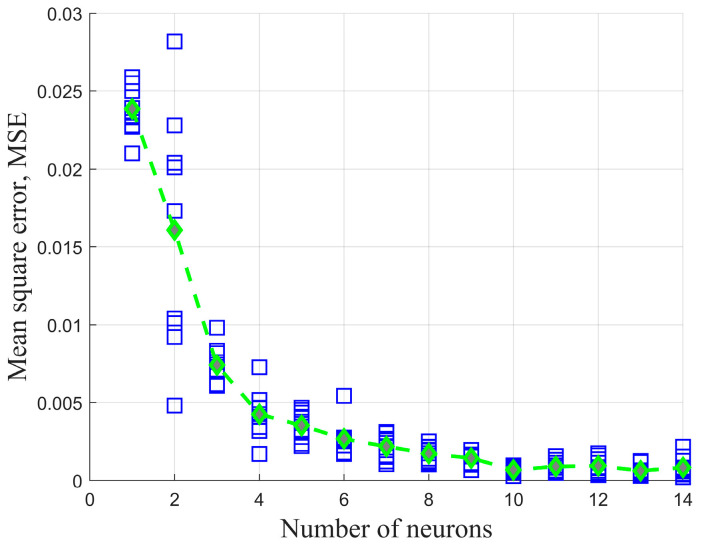
MSE convergence results of BP neural network model.

**Figure 8 sensors-23-04533-f008:**
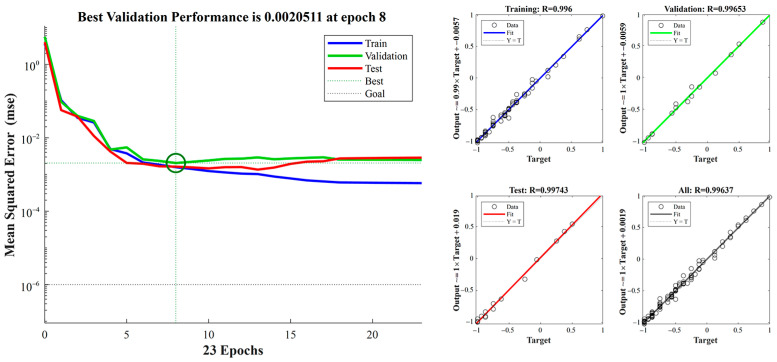
Training results. The left side is the root mean square error graph corresponding to the training set, verification set, and test set, and the right side is the corresponding correlation coefficient, respectively.

**Figure 9 sensors-23-04533-f009:**
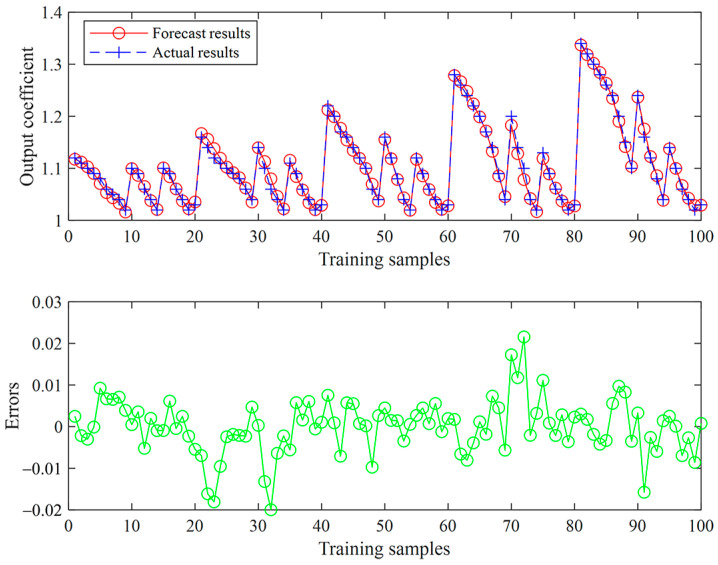
Error of the training results.

**Figure 10 sensors-23-04533-f010:**
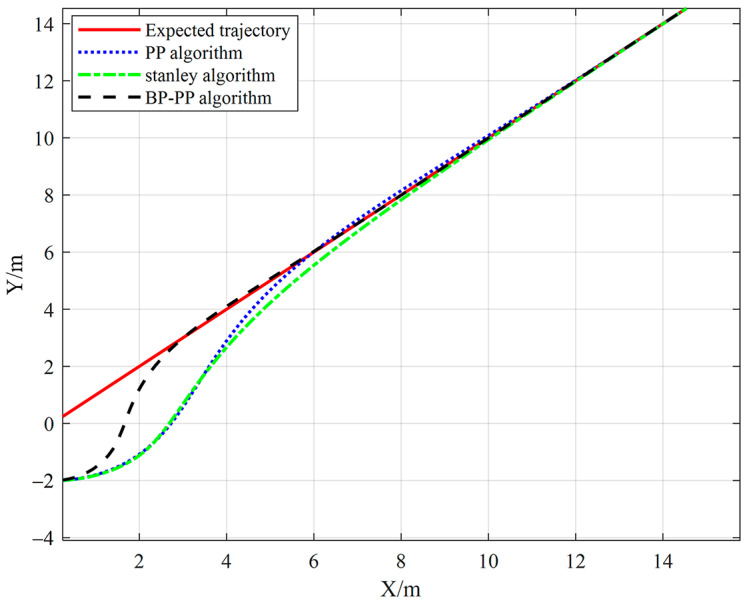
Straight tracking results.

**Figure 11 sensors-23-04533-f011:**
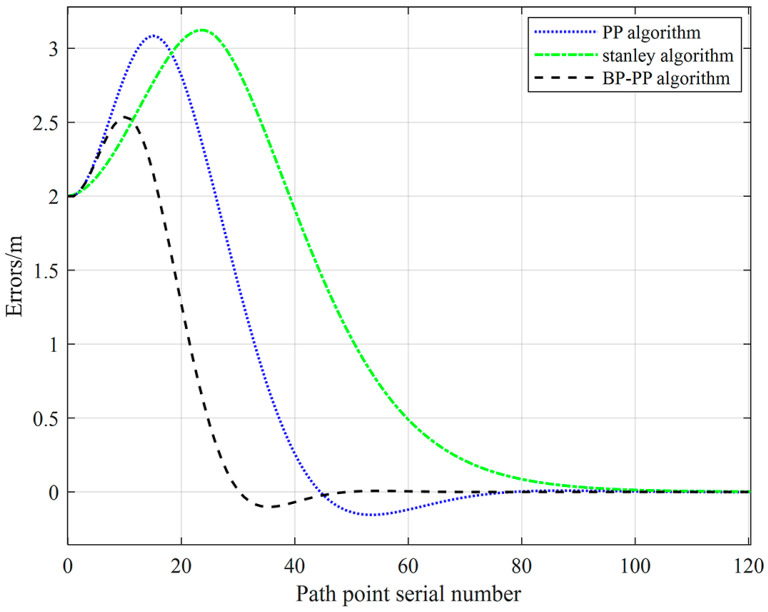
Straight path tracking error.

**Figure 12 sensors-23-04533-f012:**
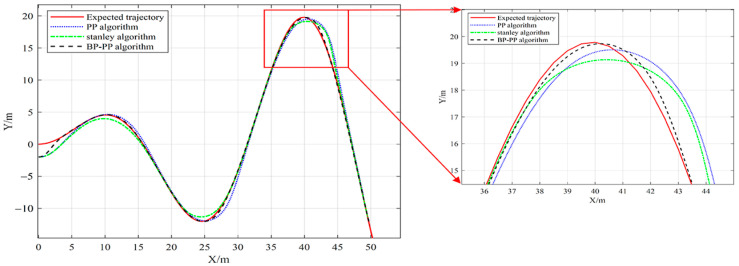
Curve tracking results.

**Figure 13 sensors-23-04533-f013:**
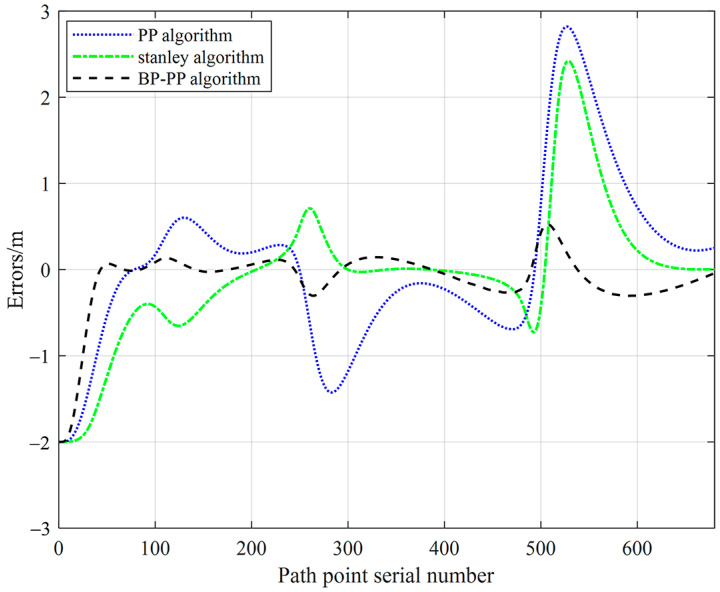
Curve tracking error.

**Figure 14 sensors-23-04533-f014:**
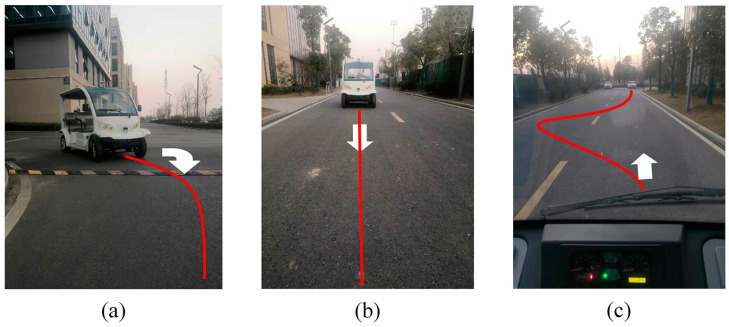
(**a**) Right angle curve; (**b**) Straight path; (**c**) Large curvature path.

**Figure 15 sensors-23-04533-f015:**
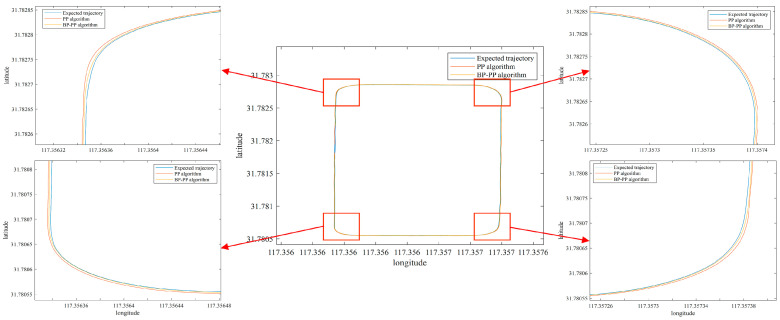
Rectangular path tracking.

**Figure 16 sensors-23-04533-f016:**
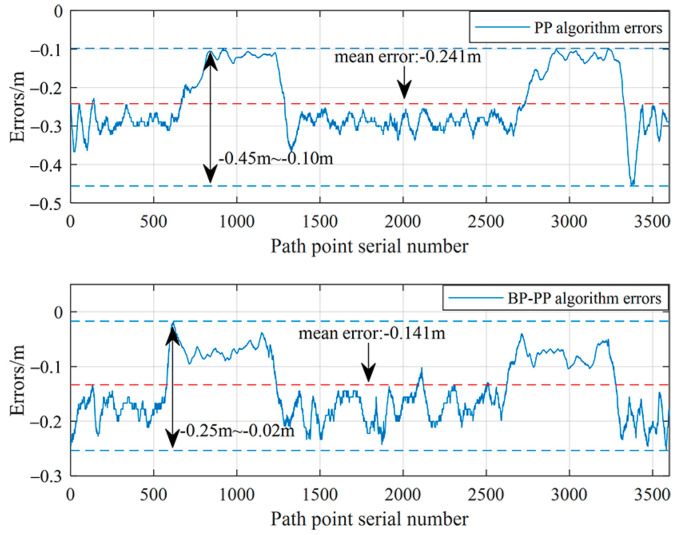
Comparison error between the PP and BP–PP algorithms.

**Figure 17 sensors-23-04533-f017:**
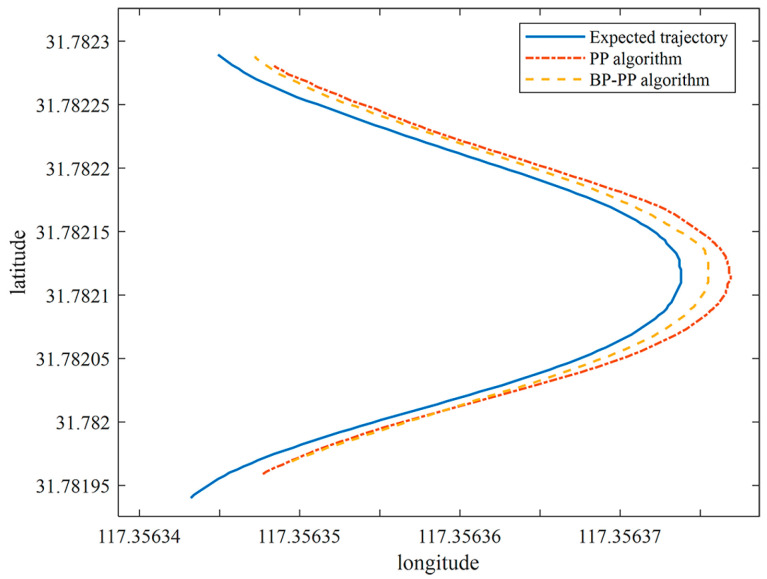
Comparison of the large curvature tracking.

**Figure 18 sensors-23-04533-f018:**
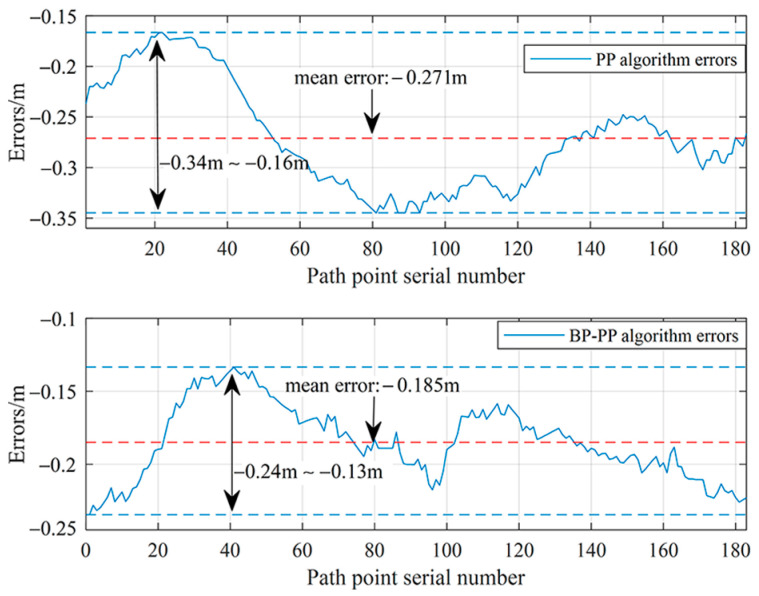
Comparing lateral errors between the PP algorithm and the BP–PP algorithm.

**Table 1 sensors-23-04533-t001:** Main parameters of the electric patrol vehicle.

Parameters	Value
Vehicle type	electric wheeled vehicle
Steering mode	front wheel steering
Maximum turn	−30°∼+30°
Wheelbase	2.3 m
Track	1.2 m
Weight	800 kg
Size $\left(l\times w\times h\right)$	$3.1\times 1.44\times 1.95\left(m\times m\times m\right)$

**Table 2 sensors-23-04533-t002:** Results of the tracking error.

Path Section	Average Curvature Radius (m)	Average Error PP (m)	Average Error Stanley (m)	Average Error BP–PP (m)
1	6.80	0.31	0.53	0.04
2	5.50	0.52	0.14	0.10
3	4.50	0.86	0.47	0.19

**Table 3 sensors-23-04533-t003:** Rectangular path tracking data.

Algorithm	Initial Error (m)	Max Error (m)	Mean Error (m)
PP	−0.240	−0.450	−0.241
BP–PP	−0.240	−0.250	−0.141

**Table 4 sensors-23-04533-t004:** Arch path tracking data.

Algorithm	Initial Error (m)	Max Error (m)	Mean Error (m)
PP	−0.240	−0.340	−0.271
BP–PP	−0.240	−0.240	−0.185

## Data Availability

Not applicable.

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
