# Peer review of "Investigating the Path Tracking Algorithm Based on BP Neural Network"

_sensors, 2023, doi:10.3390/s23094533_

Round 1

Reviewer 1 Report

The topic undertaken by the authors is very interesting. However, the article contains some errors and shortcomings that should be taken into account before publication. 

1. In the abstract, it would be worth mentioning what practical application? commercial? for whom? may have an algorithm proposed by the authors.

2. At the end of the Introduction section, there is no detailed description of the aim of the work.

3. There is also no clear indication of the originality and novelty of the conducted research.

4. At the end of the Introduction section, there is also no description of the organization of the entire manuscript - that is, what the individual sections contain.

5. In my opinion, due to the lack of a Literature Review section, the Introduction section should definitely be expanded by adding more theoretical information, along with providing references on which later research is based.

6. There is no information about whose property is an electric patrol vehicle, which is the control object in this paper.

7. Was the diagram in figure 2 developed by the authors themselves? Or has it been developed based on any literature sources? The same also applies to Figures 3 and 4.

8. There is no detailed explanation, and above all, no discussion of the results obtained and confronting them with the results obtained by other authors in this area.

9. The article was based on too few references; moreover, the vast majority - looking at the names - are articles by Chinese researchers. I think that it would be worth expanding the literature also with the works of authors from other countries, eg European or American.

Reviewer 2 Report

In this paper, we propose an adaptive path tracking algorithm based on BP (back propagation) neural network to increase the performance of vehicle path tracking in different environments. Specifically, based on the kinematic model of the vehicle, the front wheel steering angle of the vehicle was derived with the PP (Pure pursuit) algorithm and related parameters affecting path tracking accuracy were analyzed.

(1) The abstract should be improved. Your point is your own work that should be further highlighted.

(2)The parameters in expressions are given and explained.

(3) The method in the context of the proposed work should be written in detail

(4) The values of parameters could be a complicated problem itself, how the authors give the values of parameters in the used methods.

(5) The literature review is poor in this paper. You must review all significant similar works that have been done. I hope that the authors can add some new references in order to improve the reviews and the connection with the literatures. 10.1088/1361-6501/ac9a61 and so on.

(6) In Line 204, "the number of training times, the training target, and the learning rate are set to 1000, 6 1.0 1 , and 0.05, respectively". How to determine thses values?

(7) Some new algorithms should be selected to compare.

(8) The authors are requested to correct all spelling mistakes.

Reviewer 3 Report

This work presents an interesting study on an adaptive path-tracking algorithm to improve the performance of an autonomous vehicle using pure-pursuit backpropagation neural network (PP-BP). Three input parameters, namely instantaneous vehicle speed, the radius of the curvature, and the lateral error, are considered in the training of BP neural networks. The control coefficient is the output parameter. Simulation and experimental results are presented to show the effectiveness of the proposed technique for different paths, such as straight, curvature, and filleted rectangular paths. The contributions of the work are fair, timely, and relevant to researchers working in the domain of path planning of autonomous vehicles. However, after carefully reading the manuscript, a few significant concerns are observed that need to be addressed before the publication:

1. Authors are suggested to highlight the contributions of the present work more clearly (point-wise) at the end of the Introduction section. Moreover,  "different environmental conditions" should be replaced with "different paths" as the environmental condition of the road is assumed to be smooth in this work.

2. The hardware navigation board with Global Navigation Satellite System (GNSS) positioning and Inertial Measurement Unit (IMU) should be shown separately.

3. The representation of \theta, the heading angle of the vehicle, is missing. Please provide a schematic illustration for the same. Moreover, all used notations should be thoroughly checked to improve the paper's readability. For example, 'x' and 'y' in Eq. 6 look similar to Eq. 9-12.

4. In the BP neural networks model, authors are strongly recommended to show a schematic diagram with an input layer, hidden layer, and output layer with used activation functions to improve the interpretability of the work.

5. What is the basis of Eq. 8? How have authors found this? There should be proper reasoning with appropriate references. Moreover, the hidden nodes (starting from 1) should be checked for the minimum converging root mean square error and then need to be fixed. Please refer to "2022. Biomechanical study and prediction of lower extremity joint movements using bayesian regularization-based backpropagation neural network." There should be adequate information on the number of nodes in the hidden layer. 

6. Authors are suggested to provide the range of input and output parameters quantitatively in Section 3.2. Moreover, the authors are meant to show the brief details of the gradient descent algorithm and involved hyperparameters. The information on the division criteria of training, testing, and validation set should be presented.

Round 2

Reviewer 2 Report

I have appreciated the deep revision of the contents and the present form of this manuscript. All my previous concerns have been accurately addressed. I think that this paper can be accepted.

Reviewer 3 Report

The authors have addressed all the concerns raised by the reviewer and the manuscript is now in good scientific rigor.